# Fire Impacts on Water Resources: A Remote Sensing Methodological Proposal for the Brazilian Cerrado

Gustavo Willy Nagel [1,*], Lino Augusto Sander De Carvalho [2], Renata Libonati [2], Andressa Karen da Silva Nemirovsky [2] and Mercedes Maria da Cunha Bustamante [3]

1    School of Ocean and Earth Science, University of Southampton, Southampton SO17 1BJ, UK
2    Department of Meteorology, Federal University of Rio de Janeiro (UFRJ), Rio de Janeiro 21941-916, RJ, Brazil
3    Department of Ecology, Institute of Biology, University of Brasília, Brasília 70910-900, DF, Brazil
*    Correspondence: gwn1n21@soton.ac.uk

**Abstract:** Fire events are increasing in frequency, duration, and severity worldwide. The combination of ash and uncovered land might increase the transportation of pollutants into the streams, potentially affecting the water supply systems. The intensifying fires in Brazil's Cerrado biome, responsible for 70% of the country's water supply, give rise to profound ecological, climatic, and socio-economic concerns that require urgent and effective mitigation strategies. However, little attention has been paid to the consequences of fire events on water resources in the region. In this study, the Fire Impact on Water Resources Index (FIWRI) is proposed and applied in six different water supply watersheds to analyse fire behaviour from 2003 to 2020 and its potential impact on inland water bodies. This is the first remote-sensing-based index for fire impact on water resources developed for the Brazilian territory, to support water management on a watershed scale and uses variables such as terrain slope, river proximity, and vegetation to classify fire events as having a low to high potential to contaminate water bodies. We observed that all six water supply watersheds suffered frequent fire events, with different FIWRI proportions, which ranged from High to Low FIWRI. The proposed index could be used in real-time fire monitoring alert systems in order to support water supply management.

**Keywords:** water resources; water supply; wildfire; remote sensing; water quality

## 1. Introduction

The vast majority of Brazilian biomes are currently subjected to burning [1]. In the Cerrado biome (savannas of central Brazil), for example, although fire has been naturally present in this environment for millennia [2], the predominance of fires today is caused by humans, either for agricultural or pasture expansion or to manage agriculture by burning residues [3]. In addition, the increased frequency and intensity of droughts, potentially linked to climate change, is another factor that raises the risk of fire, especially in the Cerrado biome [4,5]. This was corroborated by Silva [6], who identified that large fire frequency has been increasing since 2001 in many regions of the Cerrado biome. These burned areas have the potential to increase erosion and pollution load into water bodies, affecting public water distributions [7,8]. As result, considering that the Cerrado biome supplies 70% of the freshwater for the country [9], the increase in wildfires in Brazil exposes a risk of the the impact of these events on water quality and, consequently, on public water supplies, whichis still little explored in the literature.

During fire events, heat creates a layer of hydrophobic soils, which increases runoff and thus erosion rates along the watershed [8]. As a result, the first rain events after the burn event can considerably increase the transport of sediments and ash into the rivers and streams, possibly increasing the concentration of nutrients, carbon, turbidity, and heavy metals in the water [10]. Therefore, with a higher runoff, the chemical substances produced during burn events are easily eroded into streams, impacting the water quality,

the aquatic ecosystem, and the water supply for the population [7,8,10]. During burning events, ash, a particulate residue composed of minerals and compound substances, is produced according to combustion completeness, ranging from black ash (incomplete combustion), rich in charred organic material, and white ash (complete combustion), rich in inorganic nutrients such as ammonium, nitrate, soluble reactive phosphate (SRP), and potassium [11–13]. These components might produce toxic environments and disrupt aquatic ecosystems, as found by Gonino [14], which identified that the sugarcane burning in Brazil negatively affected fish species. Furthermore, heavy metals and trace elements are also released by vegetation and organic matter through combustion or the interactions of fire and soil which induce ash and charcoal deposition [15,16]. These substances can be transported into the streams and cause severe health problems in the populations that are supplied by these water bodies [15,17,18].

However, the duration and intensity of this transport varies immensely according to watershed and climate characteristics [19]. The amount of ash, the degree of terrain slope, the rain intensity, and the proximity to streams (riparian zones) are variables that might promote a more intense and immediate fire impact on water resources [20,21]. Brito [22], for example, using a laboratory approach, found that the ash produced in the Cerrado biome has the potential to increase the dissolved solids and conductivity levels, increase the pH and decrease the dissolved oxygen. These consequences negatively impact water supply operations, increasing treatment costs and affecting the population's health [23]. Smith [17] informed that the post-fire damage to the water supply will depend mainly on the water treatment capacity, the availability of different water sources, and the size of the population that is supplied by this contaminated water body. In Brazil, where the fire frequency is increasing and the investment in water supply infrastructure has been below the recommendations by the World Bank [24], water supply disruptions have been recurrent and likely to persist in the near future [25], a potential problem that has not yet been investigated. Furthermore, in contrast to Brazil, fire impacts on water resources are well documented worldwide. The majority of studies on this topic is located in countries that suffer intense fire events in North America, Europe, and Oceania [26–31].

To monitor these fire events, satellite imagery has been used due to its systematic Earth coverage, enabling fire spatial and temporal analyses [32]. These include fire detection, fuel estimation, fire risk mapping, post-fire vegetation recovery monitoring, and even real-time fire monitoring [33–35]. Pinto [35], for example, developed a methodology to detect burning areas using deep learning that is now part of the ALARMES (https://alarmes.lasa.ufrj.br/ (accessed on 1 May 2023)) real-time fire monitoring (from Portuguese: Alarm system of Satellite-derived Burned Area Estimations), which is widely used by the government to aid in the management of burned areas across the Cerrado biome. However, the use of remote sensing to monitor fire impact on water resources is scarce. Robinne [36], for example, used different variables such as burned area, soil moisture, surface runoff, and evapotranspiration to create a global index to identify the water resources exposed to wildfires. On the regional scale, Robinne [37] developed the Source Exposure Index (SEI) that uses water demand information, fire activity, forest cover, source of water origin, and source water volume to identify the watersheds that are more exposed to fire impact on water resources in a region of Canada. Furthermore, Yang [38] developed a remote sensing index that uses the Revised Universal Soil Loss Equation (RUSLE) parameters and cloud computing to assess the erosion risk after fire events in a Sydney drinking water watershed. However, since RUSLE was specifically developed for assessing soil erosion, it may underestimate the true impact on water resources by not accounting for the high susceptibility of easily eroded ash particles. As a result, a remote-sensing-based index that is suitable for near real-time fire monitoring impact on water resources is needed to improve water management, especially in fire-prone regions such as the Brazilian Cerrado Biome.

The objective of this paper was to analyse fire behaviour in important watersheds located in the Cerrado biome region in order to propose an easy-to-implement remote-sensing-based index that will be included in the ALARMES to estimate fire impact on water

resources at a watershed scale. The method is based on the Fire Impact on Water Resources Index (FIWRI), which uses variables such as land cover, slope, and stream proximity based on remote sensing data, to classify fires from low to high water impact. We applied the index to analyse fire events in different Cerrado water supply watersheds in Brazilian cities including Palmas (TO), Brasília (DF), Cuiabá (MT), Goiânia (GO), Barreiras (BA), and Balsas (MA), using Google Earth Engine (GEE) cloud computing. For this, the MODIS fire product [39], with a 500 m spatial resolution, was used to investigate the spatial and temporal variation of burned areas between 2003 and 2020, years which have available MODIS data. The scalability of ALARMES in Brazil presents a unique opportunity to test the FIWRI on a national scale, thereby encouraging water managers to collect pre- and post-fire water quality data. This widespread implementation will enable the calibration and validation of the index, ensuring its alignment with specific objectives and purposes related to water management.

## 2. Materials and Methods

### 2.1. Study Area

We evaluated fire data within the main water catchments of important Brazilian Cerrado cities with large population such as Palmas, with 313,349 inhabitants in 2020 according to IBGE [40], Brasília (3,094,325 inhabitants), Cuiabá (623,614 inhabitants), Goiânia (1,555,626 inhabitants), and cities with smaller population such as Barreiras (158,432 inhabitants) and Balsas (96,951 inhabitants). All these cities, located in different areas of the Cerrado biome have the potential to be impacted by frequent fire events [3]. Silva [6] has provided an analysis of the contemporary fire patterns in Cerrado from 2001 to 2019 and showed that Cuiabá is located in an ecoregion characterized by a fire regime of moderate amounts of burned area and fire intensity, Brasília and Goiânia have a moderate burned area and a low fire intensity, Palmas and Balsas have a high burned area and a moderate fire intensity, and Barreiras has a high burned area and fire intensity. To identify the water supply watersheds in these cities, we first detected the location of the drinking water extraction areas that provide the greatest flow for the public supply of each city, according to data provided by the Brazilian National Water Agency [41]. For the city of Cuiabá (MT), a water extraction region was considered where the drainage basin encompasses other water extraction regions. For this case, the water production of all catchment points was added. Then, the drainage network was created for all studied watersheds using the digital model elevation derived from the Shuttle Radar Topography Mission (SRTM) and processing tools available in ArcMap [42] and GEE. The water supply watershed regions and their characteristics are presented in Figure 1.

### 2.1.1. Fire Occurrence

The defined water supply watersheds were uploaded into the Google Earth Engine (GEE) system and used to delimit the burn analysis area within the GEE (Figure 2). With this product available in GEE, it was possible to analyse the occurrence of fire through space and time in the selected watersheds, from 2003 to 2020 using the MODIS Terra and Aqua burned area product, namely MCD64A1 c6 [43]. This MODIS fire product comes from MODIS surface reflectance imagery with a 500 m spatial resolution and is estimated using observations of active fires using a burn-sensitive vegetation index (VI) applied to MODIS imagery with a 1000 m spatial resolution [43]. Larger uncertainties are observed in the MCD64A1 product over the southern region of Cerrado, which is in line with the presence of small and fragmented fire scars caused by pasture and croplands. Conversely, uncertainties in the north are generally low, owing to the prevalence of larger fire patches [44,45]. The burned area pixels of all the images from the analysed period were summed to generate a map of fire occurrence within the studied basins (Figure 2(3)). Furthermore, the burning events were grouped annually from the sum of all the pixels to analyse the burned areas' temporal variation in relative terms (in relation to the basin area)

(Figure 2(4)), and the fires' monthly variability by dividing the sum of fires of each month by the total number of fires (Figure 2(5)).

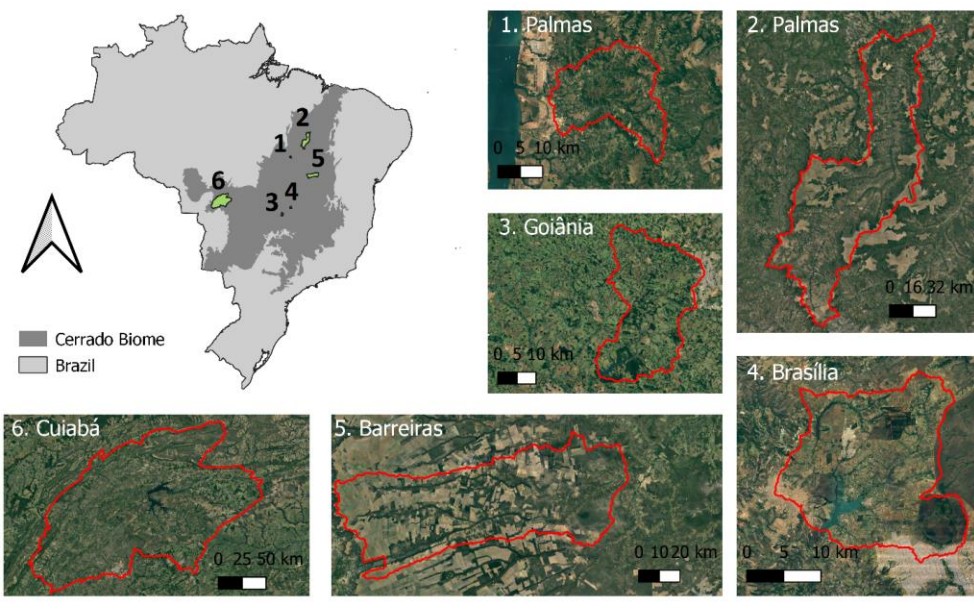

| City | State | Main River | Abstraction Discharge (l/s) | Watershed Area (ha) |
|---|---|---|---|---|
| Palmas | TO | Stream Taquarussu | 330 | 40,079 |
| Brasília | DF | River Descoberto | 4000 | 43,357 |
| Cuiabá | MT | River Cuiabá | 1837 | 2,467,181 |
| Goiânia | GO | Stream João Leite | 1733 | 70,025 |
| Barreiras | BA | River Ondas | 325 | 551,091 |
| Balsas | MA | River Balsas | 105 | 905,082 |

**Figure 1.** Study area and watershed characteristics (location, main river, discharge, and area) in six Brazilian cities within the Cerrado biome (dark grey in the main map).

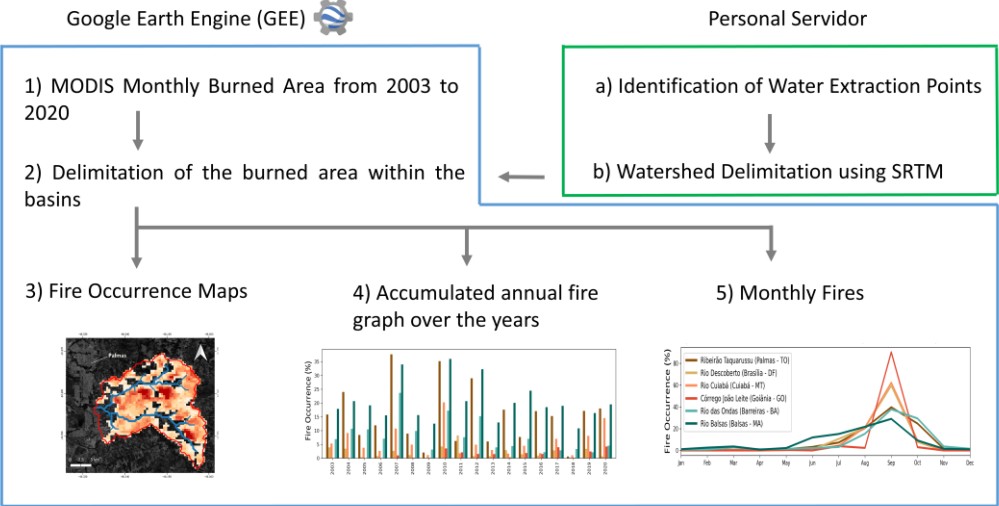

**Figure 2.** Flowchart of the burned area methodology in watersheds.

### 2.1.2. Fire Impact on Water Resource Index (FIWRI)

To identify fire patterns and their potential impact on water resources we used different datasets related to fire occurrence and watershed morphological characteristics, such as terrain slope, drainage network, and land cover (Table 1). The watershed morphological characteristics were extracted from different sources. The land cover was derived from Mapbiomas maps, from 2003 to 2019 (https://mapbiomas.org/en (accessed on 15 May 2022)) [46]. The monthly burned area was obtained from the MODIS Terra and Aqua sensor, namely MCD64A1 c6 [43], from 2003 to 2020. The network drainage and terrain slope were obtained from SRTM using hydrological tools in the ArcMap software 10.8.

**Table 1.** Databases used for the FIWRI.

| Variable | Product | Source |
|----------|---------|--------|
| Fire | MODIS burned area (MCD64A1 c6) | NASA |
| Vegetation | MapBiomas Collection 7 | MapBiomas |
| Slope | Calculated from SRTM | USGS |
| Drainage Network | Calculated from SRTM | USGS |

To estimate how fires have the potential to impact water resources, we proposed the Fire Impact on Water Resources Index (FIWRI). The FIWRI is a theoretical qualitative index that uses four different variables: fire, river proximity, slope, and land cover. The primary objective of the FIWRI is to be implemented in the areas of active burning identified by the ALARMES real-time fire monitoring system, which leverages deep learning algorithms and utilizes VIIRS Level 1B and VIIRS Active Fires data, including Fire Radiative Power (FRP), for effective fire event monitoring. However, for the purpose of this paper, we applied the FIWRI on MODIS fire areas to facilitate time-series analysis using cloud computing. By adopting this approach, our intention is to develop a practical and straightforward tool that can be readily employed to assess the potential risk of fire impact on water resources within the context of real-time fire monitoring. This methodology enables timely evaluation and monitoring of fire-related risks, contributing to enhanced decision-making and resource management strategies for mitigating the impact of fires on water resources.

We hypothesized that fires that occur close to streams have a higher probability to deliver pollutants into the streams and thus impact water quality, as mentioned by Pettit and Naiman [20]. The slope is another important variable that influences the rate of erosion because steep terrains facilitate the mobilization of fire ash pollutants into streams, leading to decreasing water quality [47]. Vegetation type was also considered as a variable since it contributes to the amount of ash released during a fire event that will be transported into streams. High biomass vegetation, such as forests, would lead to higher amounts of ash, and thus, can have a huge impact on water resources [48]. Nonetheless, low biomass vegetation, such as savanna, pasture, and agriculture, would leave the soil completely uncovered after fire events, increasing the runoff and therefore, pollutant transportation into water resources [49]. Furthermore, in regions of pasture, the fire impact might be intensified, since the hoof action of livestock will decrease the soil disaggregation, increasing soil erodibility and, consequently, impact on water quality [48]. As a result, both burned forests and low vegetation have a great potential impact on water resources.

We propose an index, based on a binary classification (fire with immediate potential impact and fire without immediate potential impact on water resources) for each of the three considered variables (terrain slope, drainage network, and land cover) (Figure 3). Fires that occur closer than 100 m from a stream are considered to have a high potential impact on water resources since the distance for the ash pollutants to reach the water resources is short. The 100 m value was considered according to the Brazilian Permanent Conservation Units (APP), which advises that rivers with a width between 50 and 200 m must have a vegetation cover within 100 m. Furthermore, fires that occur on lands with a slope above 9% are considered to have a higher probability to erode and thus have immediate potential impacts on water resources. The 9% threshold was extracted from the

Universal Soil Loss Equation (USLE), which showed that erosion is greater in lands steeper than a 9% slope [50]. Since high and low vegetation cover have a great impact on the water when burned, fires in forest, savanna, pastures, and agricultural areas were considered land covers with an immediate potential impact on water resources. The FIWRI uses these three variables to classify fires with Low FIWRI (all three variables with low impact), Low to Intermediate FIWRI (one variable with high impact), Intermediate FIWRI (two variables with high impact), and High FIWRI (all three variables with high impact) (Figure 3). We analysed the FIWRI variation temporally (considering the areas burned each year using the MODIS monthly burned area product and the Mapbiomas land cover product from 2003 to 2019) and spatially, creating a map of FIWRI with the most frequent classes (mode).

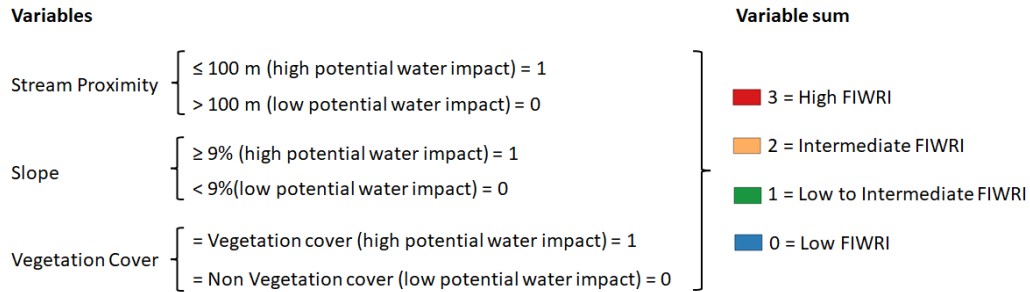

**Figure 3.** Fire Impact on Water Resources Index (FIWRI) methodology.

## 3. Results

### 3.1. Fire Occurrence Patterns in Selected Watersheds in the Brazilian Cerrado

Fires occurred in all analysed watersheds with different spatial and temporal dynamics within the Cerrado biome (Figure 4). The Taquarussu Watershed (Palmas, TO), for example, experienced fire in almost all regions at least one time from 2003 to 2020, with some of them reaching a maximum occurrence of 13 times during the analysed period. The Descoberto River (Brasília—DF) and João Leite stream (Goiânia—GO) experienced fires in more specific locations, with regions reaching a maximum of six fire events in both watersheds. It is interesting to note that in the João Leite Stream Watershed, the fire accumulated in the south, close to the reservoir used to supply the Goiânia capital. The Cuiabá River (Cuiabá—MT), Ondas River (Barreiras—BA), and Balsas River (Balsas—BA) are larger watersheds (see Figure 1) with widespread fire events, with regions reaching a maximum of 13 events in the Cuiabá River, 8 in Ondas River, and a staggering 19 fire events in Balsas River Watersheds. The Balsas River Watershed suffered the most from fire, especially in the head of the basin (south region) (Figure 4).

Historically, it was possible to notice that in most of the analysed watersheds, fires occurred every year, consuming high proportions of the land in some basins (Figure 5a). In the Taquarussu Watershed, for example, fire consumed an average of 15.5% of the basin land per year between 2003 and 2020, with some years exceeding 30%, such as in 2007 and 2010 (Figure 5a). The Balsas River is the most fire-frequent basin, with 20.3% of the land burned on average, with years reaching 34% (2007), 36% (2010), and 32% (2012) of the basin (Figure 5a). This was followed by the Ondas River Watershed with 8% of the watershed burned every year on average, Cuiabá River with 6%, Descoberto River with 2.2%, and João Leite stream with 14% (Figure 5a). It is important to note that the João Leite stream started to have fire events after 2007, increasing its burned area ever since. Despite the slight variability in the monthly fire distribution, burned areas in these watersheds occurred mainly between August and October, reaching a maximum in September (dry season in the Brazilian biome) (Figure 5b). The Balsas River Watershed, for example, had the most distributed fires throughout the year (all months experienced at least one fire event between 2003 and 2020), while the João Leite stream was the watershed with the most concentrated fires, with 90% of the fire events occurring in September (Figure 5b).

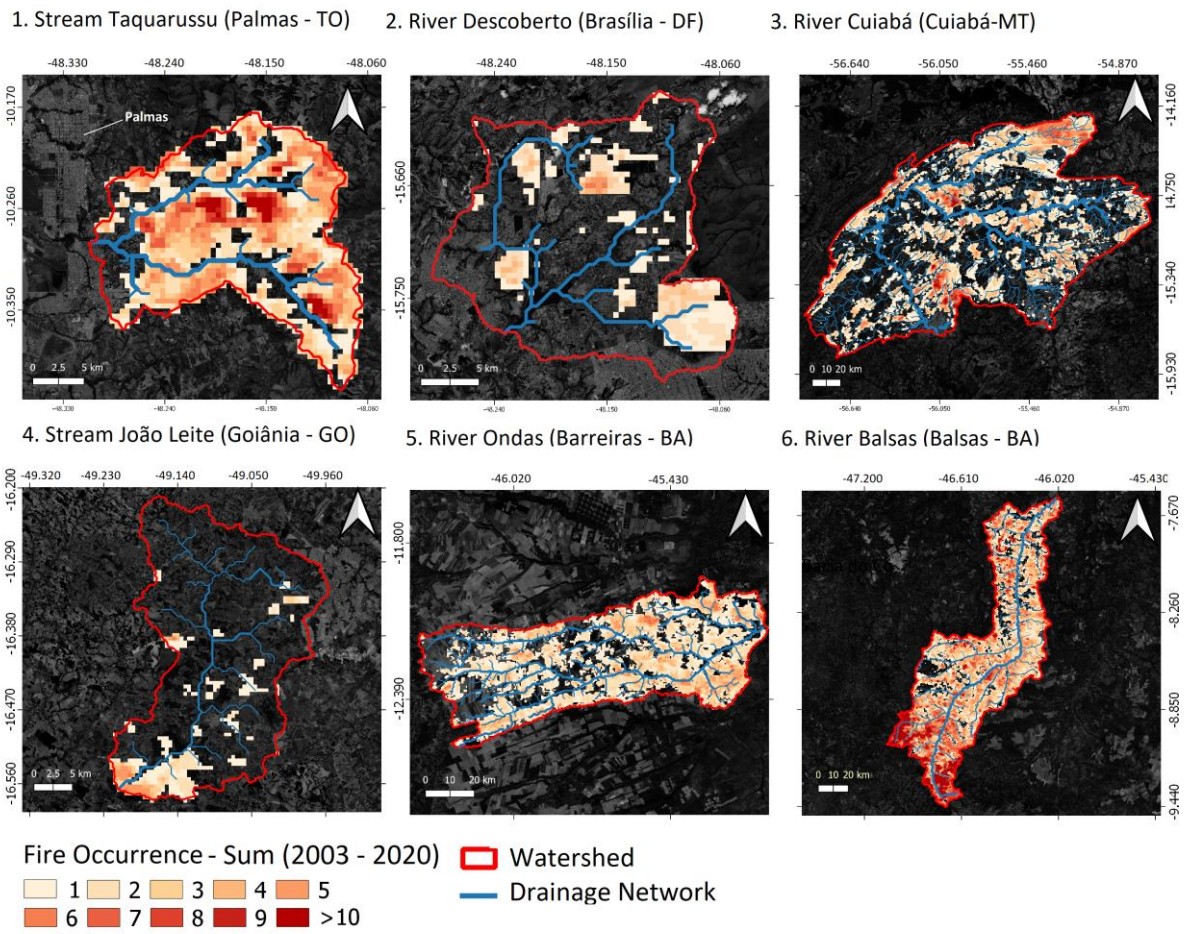

**Figure 4.** Water supply watersheds and their fire occurrence between 2003 and 2020.

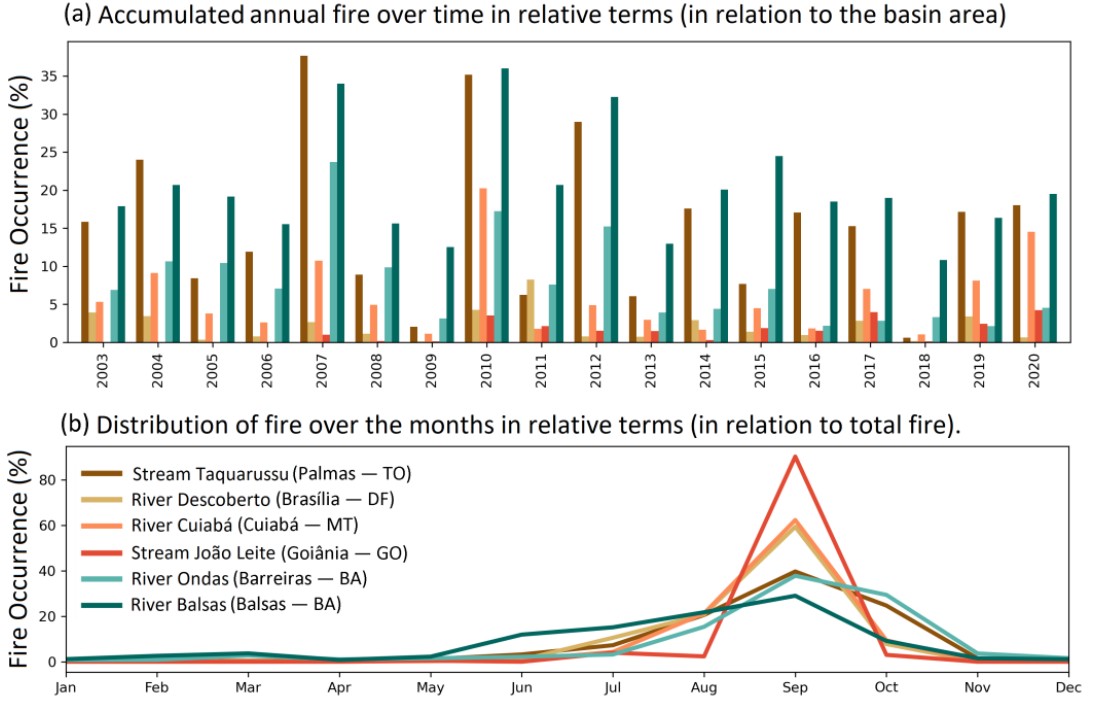

**Figure 5.** (**a**) Variation of accumulated annual fire over time in relative terms (in relation to the basin area). (**b**) Distribution of fire over the year in relative terms (in relation to total fire).

### 3.2. Fire Impact on Water Resources Index (FIWRI)

Overall, the majority of the classified fires in all analysed watersheds had Low-to-Intermediate and the Intermediate FIWRI scores, while watersheds with the extremes, Low and High FIWRI, were the minority (Figure 6). This means that most of the fires had at least one FIWRI variable with potential water impact (Low-to-Intermediate FIWRI), which might be the vegetation cover, slope, or river proximity, or a combination of two variables (Intermediate FIWRI). Although still lower in proportion, areas with a High FIWRI had a higher occurrence in the João Leite stream Watershed, with 10% on average (Figure 6B(d)). This can be explained by the fact that this watershed has steeper slopes (on average 9%), a high number of rivers, and vegetation coverage in the burned southern part of the basin (Figure 4(4)). The watershed with the next highest occurrence of High FIWRI was the Descoberto River (with 4% on average), followed by Taquarussu stream (2%), Cuiabá River (2%), Balsas River (2%), and Ondas River (0.3%). The Ondas River is a relatively flat watershed, with a 2% slope on average, which explains its low proportion of High FIWRI areas. It is possible to notice that the smaller the watershed, the higher the FIWRI annual variability. Small watersheds, such as the Taquarussu stream, Descoberto River, and João Leite stream, have a higher FIWRI variability than large watersheds, such as the Ondas River, Cuiabá River, and Balsas River (Figure 6B(b)). This is expected since in large watersheds the average FIWRI will not be highly affected by individual fires that happen in low or high FIWRI areas. As a result, the FIWRI extremes were less common (in proportion) in large watersheds, where the classes of Low-to-Intermediate and Intermediate FIWRI prevailed (Figure 6B(b)). By contrast, in small watersheds, periods with low fire extension had a higher Low FIWRI predominance (Figure 6B(b)), such as in 2009 (Figure 5b). The João Leite Stream Watershed had no fire events before 2007.

**Figure 6.** *Cont.*

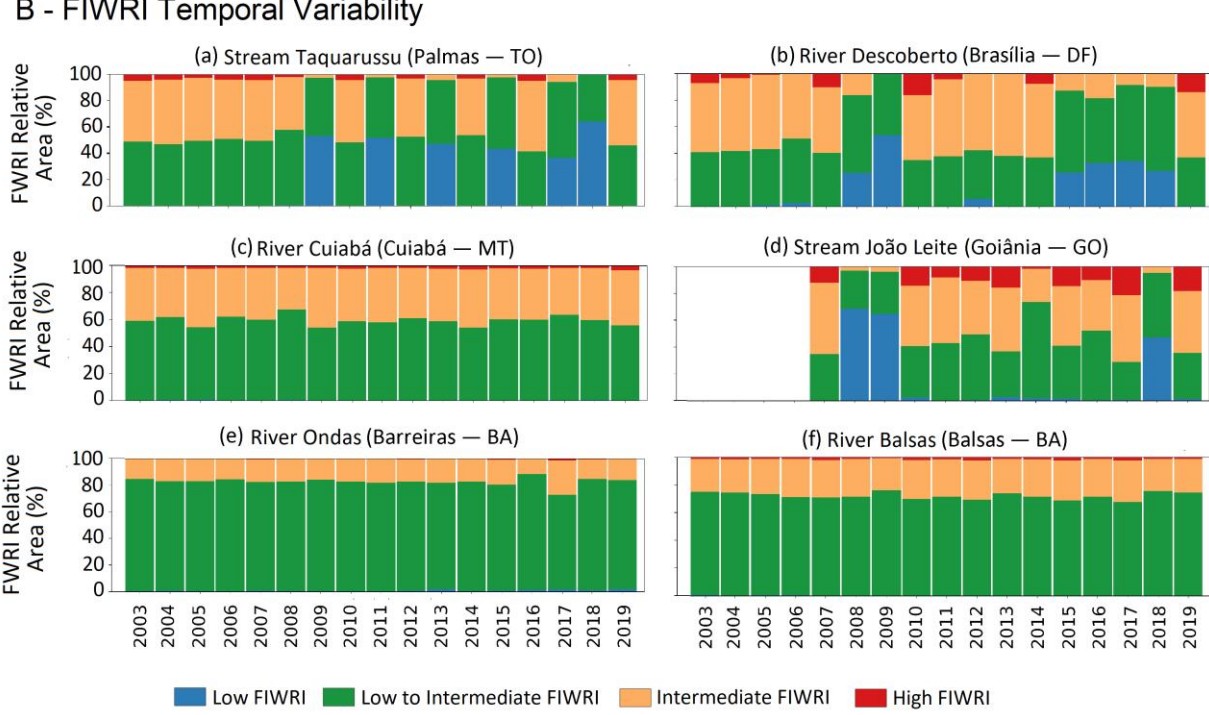

**Figure 6.** Spatial distribution of the most frequent FIWRI classifications from 2003 to 2019 for the studied watersheds (**A**), and the proportion of FIWRI by year for each watershed (**B**).

## 4. Discussion

We showed that all the six studied water supply watersheds suffered periodic fire events in their territory with different Fire Impact on Water Resources Index (FIWRI) proportions. The FIRWI is an easy-to-implement index that uses only remote sensing data to classify fires according to their water impact using variables such as river proximity, terrain slope, and vegetation cover. This straightforward method can be implemented in near real-time fire monitoring or to analyse specific fire events in water supply watersheds. So far, indexes were being developed to classify regions and watersheds with high risks of water disruption caused by fires [36,37], such as the Source Exposure Index (SEI) [37]. The FIRWI, as a result, might be further investigated for application in real-time monitoring of watersheds classified as having a high water disruption risk related to fires (using indexes developed for this purpose), in order to improve operational water management. We assumed that fires in regions close to a river, in steep terrains, and with vegetation cover will produce a higher impact on the water quality. Considering that fires impact water quality for at least a year after the burning event [29,31], frequent and systematic burning in some Brazilian watersheds might accumulate these impacts and prevent water quality recovery. To address this problem, more investigation and monitoring of the relationship between fires and the water quality in Brazil are necessary, especially in the fire-prone Cerrado biome. This knowledge will help to support decision making to improve the resilience of the Brazilian water systems and prevent the population from developing health-related problems.

On average, we found that the majority of fires were classified as Intermediate and Low-to-Intermediate FIWRI in the studied watersheds. These FIWRI class differences and their real impact on water resources must be further investigated. Furthermore, we found a higher FIWRI variability and higher High FIWRI presence throughout the study period in smaller watersheds. For large watersheds, such as the Cuiabá River, Ondas River, and Balsas River, the FIWRI proportion was more consistent over time. This was expected since we combined the fires over an entire year of large regions. However, the great advantage of FIWRI, not explored in this article, is that it can be applied to individual fire events

to determine the potential impact on water resources, helping the water management authorities implement near real-time responses. A wildfire event with a high FIWRI, for example, might give the authorities enough advanced notice to adapt the water supply system for the higher load of pollutants, or even search for new temporary water sources, improving the resilience of water supply systems in the Cerrado Biome.

Nevertheless, the vegetation removal caused by fire events decreases water interception and thus creates greater runoff even with lower intensity storms [7,8,19,51]. It is estimated that unburned shrubs can intercept more than 60% of low-intensity rainfall (13 mm over a 19 h period) and 20% during high-intensity rainfall (>70 mm over a 15 h period) [52], while rangeland plants are able to reduce erosion by 50% [53]. These soil protections are lost after severe fire events. As a result, the ash is easily eroded into streams when rainstorms occur in burned areas, elevating nutrient concentrations and possible stream temperature [54]. These conditions are ideal for algal blooms in reservoirs, which in turn release harmful toxins and strain water treatment operations [55–59]. The post-fire higher runoff and erosion rates also increase the downhill sediment transportation into the streams, affecting the water quality and the aquatic ecology [51]. The higher concentration of the total suspended sediments (TSS) in streams and lakes limits light penetration, negatively impacting primary production and thus the entire aquatic ecosystem [17,19,31,51,60]. Furthermore, fires might change the soil's physical and chemical properties and create a layer of soil water repellency (SWR) that decreases water infiltration and thus increases runoff and erosion rates even further [8]. Rodrigues [61], using a hydrological model to study a watershed that provides water to a large Brazilian urban agglomeration (Belo Horizonte), found that the loss of vegetation cover reduced infiltration, which compromised groundwater recharge. As a result, although the FIWRI was developed for water quality-related impacts, it might also be adapted to groundwater impacts, including variables such as the locations of the groundwater recharge.

The FIWRI was originally designed for integration into the near real-time fire monitoring system (called ALARMES—https://alarmes.lasa.ufrj.br/ (accessed on 1 May 2023) which tracks the daily evolution of burned areas throughout Brazil. This system is currently being used by firefighters for planning and management and is relevant to support various types of emergency management activities and to the assessment of threats to life, property, and natural resources. An important application is the planning of suppression operations during severe outbreaks of fires lasting multiple days. As the ALARMES system currently reports the daily evolution of burned areas and not severity, our initial goal is to apply the FIWRI to the ALARMES final burned product. While the currently proposed index only considers burned areas for the index implementation, we believe that the FIWRI's flexible approach could easily accommodate additional parameters, including burn severity, based on the specific watershed conditions and goals of the end-users (such as water quality for water supply or ecology). By incorporating such variables, the FIWRI can be tailored to meet the unique needs of various stakeholders, making it an even more valuable tool for fire management and mitigation efforts.

We were unable to collect in situ water quality data for the analysed watersheds before and after the wildfires and storm events. However, the primary aim of our article was to propose a remote sensing index for assessing water quality impact due to fire events. We did a survey of remote sensing-based fire assessment articles, which revealed a nota-ble disparity between the North and South Hemispheres, particularly in Brazil, a country prone to recurring fire events, where we found a limited number of papers addressing this topic, and none of them specifically utilizing remote sensing techniques to assess the im-pact of fires on water resources. (see Supplementary Material). As such, we believe that our work can serve as a catalyst for further discussion and inspire similar research efforts in other regions in Brazil, which can help address this data gap by collecting water quality data before and after fire events. Moreover, we are committed to advancing our own research efforts in this area and are actively seeking financial support to collect in situ data in various watersheds across Brazil during the fire season. We believe that such efforts

are crucial in bridging the gap between remote sensing and ground truth data and will ultimately help improve the accuracy and effectiveness of the FIWRI. Furthermore, with in situ data available, the index could be adjusted to be applied in other environments, such as Europe [62] and North America.

Despite the recognized satellite imagery capabilities, our paper is the first attempt to use remote sensing to study the potential impacts that fire has on water resources in Brazil. The index employs discrete variables, simplifying the complexity between fire and water quality. However, with in situ data, more slope and vegetation categories, and other variables, can be included to enhance its accuracy. Another index limitation is the fact that the MODIS fire product has a 500 m resolution, while the watershed characteristics, such as land cover and slope have a resolution of 30 m. This scale difference brings uncertainty to the extension of fires and thus on the relationship with watershed land cover and slope. However, despite the spatial resolution limitation, the MODIS product was essential to identifying fire behaviour over the years, from 2003 to 2020. The simple FIWRI approach means that it can be easily replicated in other studies, helping to improve the index for different environments and conditions.

## 5. Conclusions

This article was an attempt to bring awareness to the risks that fire events can cause to water quality, especially in the Brazilian water supply systems. The methodology, which uses an index for the interactions between burn areas and watershed characteristics, is unique and promising for identifying fire events with a high potential water impact. The proposed theoretical FIWRI is the first fire index that uses morphological terrain characteristics to classify the fire's potential impact on streams and lakes. We showed that all watersheds suffered fire events with different FIWRI proportions throughout the study period. The index, although requiring further research, might be applied in the future to improve water management and guarantee potable water for the Brazilian population. This study found a potential impact on six different water supply watersheds in Brazil, a condition that might be shared by hundreds or thousands of watersheds in the Brazilian Cerrado biome. As a result, more studies must be performed in Brazil to prevent future water supply disruptions and potential health impacts on the Brazilian population.

**Supplementary Materials:** The following supporting information can be downloaded at: https://www.mdpi.com/article/10.3390/fire6050214/s1.

**Author Contributions:** G.W.N. Conceptualization, Methodology, Formal analysis, Software, Writing—original draft. L.A.S.D.C. Conceptualization, Methodology, Writing—review and editing, Funding acquisition, Project administration. R.L. Conceptualization, Methodology, Funding acquisition, Writing—review and editing, Project administration. M.M.d.C.B. Methodology, Writing—review and editing. A.K.d.S.N. Writing—review and editing. All authors have read and agreed to the published version of the manuscript.

**Funding:** This research was funded by CEPF Cerrado (Critical Ecosystem Partnership Funding). L.A.S.C was funded by the project PROGRAMA FAPERJ JOVEM CIENTISTA DO NOSSO ESTADO—2021—E-26/201.40/2022. R.L. was funded by the project PROGRAMA FAPERJ CIENTISTA DO NOSSO ESTADO—2023—E-26/200.329/2023 and by CNPQ (grant number 311487/2021-1).

**Informed Consent Statement:** Not applicable.

**Data Availability Statement:** Not applicable.

**Conflicts of Interest:** The authors declare no conflict of interest.

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
