# Peer review of "Fire Impacts on Water Resources: A Remote Sensing Methodological Proposal for the Brazilian Cerrado"

_fire, doi:10.3390/fire6050214_

Round 1

Reviewer 1 Report (Previous Reviewer 1)

Dear authors, I consider that this paper is very well-written and the methods really interesting to assess fire and water resources. I have included some comments in the attached pdf related to some clarifications that needed to be included, some ideas to improve the readability of the figures and some suggestions.

Author Response

Response:

We really appreciated the suggestions, which helped us to improve the quality of the manuscript. We included the suggestions in our final paper.

In relation to Figure 5: The use of the scale would create make the smaller watersheds disappear from the map. This is because their size is very different. Stream Taquarussu, for example, has 40079 ha, while River Cuabá Watershed has 2467181 ha (61 times larger).

In relation to the lack of data for the Stream João Leite Watershed in Figure 7: This watershed had no fire events before 2007 (this was included in the text).

Do you think this index could be applied in Europe? This paper focuses its discussion on Brazil since we want to encourage more research there (there is a fire research gap between Europe/USA and South America – see Supplementary Material). However, we believe the index can be applied in other environments (we included this in the discussion section). 

Reviewer 2 Report (New Reviewer)

The authors use remotely sensed data to build an index of potential impact of fire on water resources in Brazil. The writing and methods are clear and easy to follow, and the tables and figures improve the understanding of the methodology. Unfortunately, I feel that while the authors have laid a nice groundwork for a geospatial analysis, the manuscript as it is written is not ready for publication. The authors are correct in their assertation that vegetation type and density, slope steepness, and distance from burned area to stream are all important variables for determining (some of) fire’s impact on a stream network. However, the authors leave out the impact of fire intensity/burn severity/vegetation consumption from their analysis. A low severity burn in a densely vegetated forest has a very different impact from a low severity savannah or agricultural burn. Similarly, a high severity burn in a forest is much more detrimental than a high severity burn in a less densely vegetated area. I can see their FIWRI index being useful only if there is a connection to these important fire variables (e.g. burn severity and vegetation consumption) to the FIWRI index they developed.

As it is written, the FIWRI index solely consists of stream proximity, slope, and vegetation cover. These are basic variables that have obvious but also simplistic associations with potential fire impact on hydrologic systems. The authors created many maps overlaying the index of these variables with burned area over a period of about 15 years. While there is value in seeing areas that are more likely to be impacted, adding a burn severity or vegetation consumption component would make a much greater impact. Since the authors are already using MODIS data, it would make sense to use something like NBR or NDVI (or dNBR, dNDVI for change detection) to enhance their index.

On page 10, line 306-307, the authors discuss further investigating the FIWRI in real time monitoring. They could do this using a predictive hydrological model that incorporates burn severity, soils, vegetation, topography (e.g., Dobre et al., https://doi.org/10.1016/j.jhydrol.2022.127776). On page 11, lines 342-350, the authors themselves discuss the impact of fire intensity on potential impact. Also on line 322, the authors say that the “real impact on water resources must be further investigated.”

I think that in order for this manuscript to be suitable for publication they must take some of their own suggestions and further investigate: how the FIWRI could be used in real-time; incorporate a fire intensity/burn severity/vegetation consumption component; do a deeper investigation into the real impact on water resources. The paper as written is a nice groundwork and additional analysis will result in a paper that is relevant and timely.   

Author Response

We fully acknowledge the reviewer's suggestion to incorporate fire severity/intensity into our index to enhance its effectiveness. However, it is important to note that the FIWRI was originally designed for integration into the near real-time fire monitoring system (https://alarmes.lasa.ufrj.br/) (called ALARMES, from Portuguese: Alarm system of Satellite-derived Burned area estimations) which tracks the daily evolution of burned area over Brazil. This system is currently being used by firefighters for planning and management and is relevant to support various types of emergency management activities and to the assessment of threats to life, property, and natural resources. An important application is the planning of suppression operations during severe outbreaks of fires lasting multiple days.    As the ALARMES system currently reports daily evolution of burned areas and not severity, our initial goal is to apply the FIWRI to the ALARMES final burned product. This explanation was included in the Discussion and Introduction section.

While our current proposal only considers burned areas for the index implementation, we believe that the FIWRI's flexible approach could easily accommodate additional parameters, including burn severity, based on the specific watershed conditions and goals of the end-users (such as water quality for water supply or ecology). By incorporating such variables, the FIWRI can be tailored to meet the unique needs of various stakeholders, making it an even more valuable tool for fire management and mitigation efforts (we included this in the discussion). As a result, although the index might be technically less advanced than current fire related water risk models available, the potential of it being applied in near-real time monitoring systems, such as the ALARMES system, which covers a huge area (biomes like Cerrado), will bring scalability and potentially higher benefits for water-fire management than other more intricate models.

Fire radiative power (FRP) retrievals obtained from active fire satellite observations can be use as a proxy of fire intensity, since it measures the radiant energy released per time unit by burning vegetation. However, active fire detection products often omit burned area patches (Roy et al., 2005), leading to underestimation of the area burned. Omission errors from active fire detection products may be due to the spatial and the temporal coverage of satellite overpasses, sensor saturation, or obscuration by clouds and smoke (Schroeder et al., 2008). The presence of thick clouds and heavy smoke layers is a major drawback in operational applications, preventing hot spot detection due to the spectral signal attenuation in the atmosphere (Fournier et al., 2001). This is especially true in the tropics during the dry season (Hilker et al., 2015), where the probability of cloud-free observations is, on average, less than 30% (Wylie et al., 2005). Accordingly, the use of fire intensity as derived by satellite in the formulation of our index is hampered by the accuracy of active fire products. The proposed methodology based on the use of burned area products is able to produce less omission in FIWRI.

We acknowledge the reviewer's concern regarding the lack of field data to support a detailed analysis of the impact of fires on water resources. However, it is important to note that the primary aim of our article is to propose a remote sensing index for assessing water quality impact due to fire events. As such, we believe that our work can serve as a catalyst for further discussion and inspire similar research efforts in other regions, which can help address this data gap by collecting water quality data before and after fire events.Moreover, we are committed to advancing our own research efforts in this area and are actively seeking financial support to collect in situ data in various watersheds across Brazil during the fire season. We believe that such efforts are crucial in bridging the gap between remote sensing and ground truth data, and will ultimately help improve the accuracy and effectiveness of our proposed index. We included this point in the Discussion section.

We did a survey of articles that assess fires using remote sensing, and found a discrepancy of papers in the North to the South Hemisphere. In Brazil, a country that suffers from periodic fire events, we found a limited number of papers, none of them using remote sensing to assess fire impact on water resources. Therefore, research on fire impacts on water resources in Brazil is behind other more fire-intense research countries, such as the USA. Remote Sensing can support to reduce this research gap by monitoring large fire prone areas, such as the Brazilian Cerrado Biome. As a result, we are confident that our article will have a positive impact on the promotion and popularisation of remote sensing for assessing water quality impact due to fire events. By encouraging further research and advocating for increased support for field data collection, we believe that we can pave the way for more effective fire management and mitigation strategies in the future. This analysis was mentioned in the discussion section, and included in the Suplementary material. 

Reviewer 3 Report (New Reviewer)

The authors develop a post-fire water resource (specifically, water quality) effect index dependent on proximity of the fire (centroid? downstream extent?) to streams, surface slope, and type of vegetative cover affected. The index is a simple binary discretization of the attributes of the index components with respect to criteria based on the literature (e.g., if general slope is greater than or equal to 9%, the slope component is quantified as equal to 1, to 0 otherwise). The index is comprised as the sum of the component parts, with an index value ranging from High (3) to Low (0). Further, the index is developed from remotely sensed products only. The authors present a helpful and informative flowchart of the methodology for computing the index in Figure 2. After some discussion of six fire-affected watersheds, the authors apply their index to the watersheds. Presumably, the index was developed as an alternative to current application of burn severity maps to assess watershed impact--as such, the index is a novel and interesting development!

However, there is no validation or application of the index for the stated objective of a water quality assessment of post-fire runoff. The validity and use of the index could be supported by application to a watershed using data expressed as actual water quality standards, if collected.  Understandably, local validation data may not exist, but there may be a few pre-/post-wildfire water quality data sets (with data that can be quantified in terms of drinking water quality standards) that may provide data for a simple validation and demonstration of the utility of the index. A demonstration of the index to actual water quality data is recommended. Because of this omission, acceptance of the manuscript after minor revision is suggested.

A minor but perplexing presentation of the data and results concerned the land cover data and results in both sections 2 and 3. The objective of the manuscript is the FIWRI. Discussing the land cover analysis and results (sections 2.2.3 and 3.3, respectively) after discussing the index (sections 2.2.2 and 3.2) detracts from the focus of the objective.

Author Response

We acknowledge the reviewer's valid concern regarding the limited availability of public field data to support a comprehensive analysis of the impact of fires on water resources. It is indeed a challenge to find a significant amount of relevant in situ water quality data. However, we would like to reiterate that our articles primarily focuses on proposing a remote sensing index for evaluating the water quality consequences of fire events. In light of this limitation, we believe that our work can still play a significant role in driving further discussion and inspiring similar research efforts in other regions. By highlighting the need for collecting water quality data before and after fire events, we hope to encourage researchers and stakeholders to address this data gap and contribute to the broader understanding of the impact of fires on water resources. Moreover, we are fully committed to advancing our own research in this area and actively seeking financial support to conduct in situ data collection in various watersheds across Brazil during the fire season. By bridging the gap between remote sensing and ground truth data, we aim to enhance the accuracy and effectiveness of our proposed index and contribute to the overall understanding of the relationship between fires and water quality. While acknowledging the current limitations, we remain optimistic that collaborative efforts and dedicated data collection initiatives will gradually address the scarcity of public field data and enable a more comprehensive assessment of the impact of fires on water resources.

We excluded the land cover analysis, and the proximity of the fire is in in relation to the river shoreline (100% water frequency overtime).

We did a survey of articles that assess fires using remote sensing, and found a discrepancy of papers in the North to the South Hemisphere. In Brazil, a country that suffers from periodic fire events, we found a limited number of papers, none of them using remote sensing to assess fire impact on water resources. Therefore, research on fire impacts on water resources in Brazil is behind other more fire-intense research countries, such as the USA. Remote Sensing can support to reduce this research gap by monitoring large fire prone areas, such as the Brazilian Cerrado Biome. As a result, we are confident that our article will have a positive impact on the promotion and popularisation of remote sensing for assessing water quality impact due to fire events. By encouraging further research and advocating for increased support for field data collection, we believe that we can pave the way for more effective fire management and mitigation strategies in the future. This analysis was mentioned in the discussion section, and included in the Suplementary material.  We excluded the land cover analysis of the article.

Reviewer 4 Report (New Reviewer)

General Comments

The authors synthesize ~20 yr of historical burn areas across an ecologically and water operationally significant area of Brazil, using three environmental factors (Stream proximity, Slope and Vegetation Cover) each with two categorical levels to establish a fire impact on water resources risk rating from zero to three.

I commend the authors on their project goals and logical methodology. The substantial issues with this study are as follows;

×        The modelling method does not quantify the resulting impacts on water resources, vegetation cover is split into a very simple ‘with’ and ‘without’ category as opposed to percentage cover, slope is broken down into very simple >9% and <9% categories. Additionally, while distance to stream is clearly a risk factor for water management actions, no justification for the categorisation of >100m and <100m is provided, for instance, with a slope of 8-15% and a large burn why would 100-200m not be high risk for erosion? These categories/questions are not clearly quantified/justified.

×        ×        There are multiple references to this study being ‘unique’ or a ‘first’ in it’s use of remote sensing to estimate fire effects on water resources, however this unfortunately far from accurate. Specifically see two methods widely used across North America and Australia, the WEP and RUSLE models (Neris, J., Santin, C., Lew, R., Robichaud, P. R., Elliot, W. J., Lewis, S. A., ... & Doerr, S. H. (2021). Designing tools to predict and mitigate impacts on water quality following the Australian 2019/2020 wildfires: Insights from Sydney's largest water supply catchment. Integrated Environmental Assessment and Management17(6), 1151-1161; Yang, X., Zhang, M., Oliveira, L., Ollivier, Q. R., Faulkner, S., & Roff, A. (2020). Rapid assessment of hillslope erosion risk after the 2019–2020 wildfires and storm events in Sydney drinking water catchment. Remote Sensing12(22), 3805.). The current focal point in this area of remote sensing science is how the ‘risk’ is defined, and more so, how the quantifiable water quality impacts can be linked to the ‘risk’ that is largely based on erosion rate and proximity to source (such as your proposed index).

×        Fire intensity not being included in the model is a large flaw, with total burn area not effective enough to inform erosion and chemical risk to water quality. A high intensity burn is of far greater risk to water quality than a low intensity burn.

×        A key takeaway from this work, is that the categories used to assess water risk are less advanced than current fire related water risk models available, and there is no quantified link to water quality change in the paper or quantified chemical composition of the vegetation being assessed (once burnt).

Specific Comments

Line (2): Capitalise ‘a’

Line(14-15): ‘sup-ply’ to ‘supplies’

Line(14):  ‘…for the ecological…’ to ‘…due to the ecological..’

Line(14): ‘climate’ to ‘climatic’

Line(15): ‘water that flows around the country’ should be reworded. i.e., 70% of the countries ‘drinking water’ or ‘natural flows’.

Line(18-20): The statement that this is ‘the first remote sensing index developed to support water management on a watershed scale’ is unfortunately not correct. Many watershed scale water management indexes are available with a variety of specified purposes in general, while in relation to fire and water quality see Neris, J., Santin, C., Lew, R., Robichaud, P. R., Elliot, W. J., Lewis, S. A., ... & Doerr, S. H. (2021). Designing tools to predict and mitigate impacts on water quality following the Australian 2019/2020 wildfires: Insights from Sydney's largest water supply catchment.

The statement needs to be re-written to highlight exactly what you believe is ‘the first’ in your study, or removed.

Line(35): ‘over the years’ state the exact time scale from the Silva study.

Line(36): ‘ a load of pollution’ to ‘pollution load’

Line(37): what is meant by ‘public water distributions’?

Line(37-40): Needs to be re-written, it does not make sense currently.

Line(59): ‘vary’ to ‘varies’

Line(74-76): ‘in this topic is’ to ‘on this topic are’,

Line(103): remove ‘likely’

Line(154): Figure ‘Monty’ to ‘Monthly’

Line(281): Figure 7. Suggest changing colour scheme (increase the variation in shades form light to dark) or introducing patterns to help determine differences. The current combination of red/green is difficult for those with colour-blindness.

Line(300-306): FIWRI is incorrectly spelt as FIRWI.

Line(318-334): This paragraph discusses many things that have no been quantified in this study, with a large amount of ambiguity around each claim, i.e., ‘”may” and “might” are used multiple times. The paragraph also references ‘the great advantage of FIWRI’, and then states that this advantage is ‘not explored in this article’. I’m hesitant to accept that FIWRI can be applied to individual fire events, with quantifiable implications on water quality, as this article does not present this as a method. Perhaps you can demonstrate this claim through additional work linking the FIRWA to local water quality data and flow rates/turbidity.

Line(336-350) This paragraph correctly highlights that there is a lack of data on the chemical composition of the vegetation types in your study area, however, this information is rather required for an informative water quality risk to be applied. Perhaps your next steps can be to collect this data.

Line(375-377): Agreed that rainfall intensity and fire intensity are necessary.

Author Response

The authors agree that our study does not quantify the resulting impacts on water resources, but rather assesses the risk of fire impact on water resources. However, it is important to note that the FIWRI was originally designed for integration into the near real-time fire monitoring system (https://alarmes.lasa.ufrj.br/) (called ALARMES, from Portuguese: Alarm system of Satellite-derived Burned area estimations) which tracks the daily evolution of burned area over Brazil. This system is currently being used by firefighters for planning and management and is relevant to support various types of emergency management activities and to the assessment of threats to life, property, and natural resources. An important application is the planning of suppression operations during severe outbreaks of fires lasting multiple days.    As the ALARMES system currently reports daily evolution of burned areas and not severity, our initial goal is to apply the FIWRI to the ALARMES final burned product. This explanation was included in the Discussion and Introduction section.

Fire radiative power (FRP) retrievals obtained from active fire satellite observations can be use as a proxy of fire intensity, since it measures the radiant energy released per time unit by burning vegetation. However, active fire detection products often omit burned area patches (Roy et al., 2005), leading to underestimation of the area burned. Omission errors from active fire detection products may be due to the spatial and the temporal coverage of satellite overpasses, sensor saturation, or obscuration by clouds and smoke (Schroeder et al., 2008). The presence of thick clouds and heavy smoke layers is a major drawback in operational applications, preventing hot spot detection due to the spectral signal attenuation in the atmosphere (Fournier et al., 2001). This is especially true in the tropics during the dry season (Hilker et al., 2015), where the probability of cloud-free observations is, on average, less than 30% (Wylie et al., 2005). Accordingly, the use of fire intensity as derived by satellite in the formulation of our index is hampered by the accuracy of active fire products. The proposed methodology based on the use of burned area products is able to produce less omission in FIWRI.

Regarding the categorization of vegetation cover and slope, we agree that percentage cover and more detailed slope categories would provide more information. However, our approach was intended to be simple and easy to implement, given the limitations of data availability. We recognize that more detailed studies may benefit from more detailed vegetation cover and slope categories. This explanation was included in the discussion section.

Regarding the distance to stream categorization, we used a conservative approach, based on the Brazilian APPS (Permanent Conservation Units), which informs that rivers with width between 50-200m must have a vegetation cover within 100m. However, this value and can be easily changed to calibrate a model once in situ data is available. We included this explanation in the method section.

While our current proposal only considers burned areas for the index implementation, we believe that the FIWRI's flexible approach could easily accommodate additional parameters, including burn severity, based on the specific watershed conditions and goals of the end-users (such as water quality for water supply or ecology). By incorporating such variables, the FIWRI can be tailored to meet the unique needs of various stakeholders, making it an even more valuable tool for fire management and mitigation efforts.

We acknowledge the reviewer's concern regarding the lack of field data to support a detailed analysis of the impact of fires on water resources. However, it is important to note that the primary aim of our article is to propose a remote sensing index for assessing water quality impact due to fire events. As such, we believe that our work can serve as a catalyst for further discussion and inspire similar research efforts in other regions, which can help address this data gap by collecting water quality data before and after fire events. This suggested article by the reviewer also suffered with the lack of in situ data: Yang, X., Zhang, M., Oliveira, L., Ollivier, Q. R., Faulkner, S., & Roff, A. (2020). Rapid assessment of hillslope erosion risk after the 2019–2020 wildfires and storm events in Sydney drinking water catchment. Remote Sensing, 12(22), 3805.). Moreover, we are committed to advancing our own research efforts in this area and are actively seeking financial support to collect in situ data in various watersheds across Brazil during the fire season. We believe that such efforts are crucial in bridging the gap between remote sensing and ground truth data, and will ultimately help improve the accuracy and effectiveness of our proposed index.

We removed the parts of the text saying the article was unique and included them in the introduction and discussion, and minor suggestions were all included. We thank, once more, the reviewer for the suggestions.

Line(37): what is meant by “public water distributions? The water public supply is whole system of collecting water from a reservoir, treating it and then distributing it. Fires might affect the water reservoirs, and thus the whole water supply system.

We did a survey of articles that assess fires using remote sensing, and found a discrepancy of papers in the North to the South Hemisphere. In Brazil, a country that suffers from periodic fire events, we found a limited number of papers, none of them using remote sensing to assess fire impact on water resources. Therefore, research on fire impacts on water resources in Brazil is behind other more fire-intense research countries, such as the USA. Remote Sensing can support to reduce this research gap by monitoring large fire prone areas, such as the Brazilian Cerrado Biome. As a result, we are confident that our article will have a positive impact on the promotion and popularisation of remote sensing for assessing water quality impact due to fire events. By encouraging further research and advocating for increased support for field data collection, we believe that we can pave the way for more effective fire management and mitigation strategies in the future. This analysis was mentioned in the discussion section, and included in the Suplementary material.  

Round 2

Reviewer 2 Report (New Reviewer)

The authors did a nice job responding to the reviewer comments and I feel like the manuscript is suitable for publication in its current form.

Reviewer 4 Report (New Reviewer)

The authors demonstrate clear understanding of the weaknesses in the study, while correctly arguing the importance of filling knowledge gaps in the area of GIS/Water Quality/Wildfire in Brazil and surrounding regions. 

This paper will solidify the methods currently used for land management in the region though peer-review, while providing a spring-board for further work. 

The literature review was a highly beneficial piece of the articles aims and justifications.

This manuscript is a resubmission of an earlier submission. The following is a list of the peer review reports and author responses from that submission.

Round 1

Reviewer 1 Report

Dear authors,

 thanks for your submission. I consider that the paper is very interesting and deserves to be published in Fire. However, it is indispensable that the authors try to connect more with the necessity to make a global review to assess one region of Brazil. I do not see the connection. It seems two papers joined. I included some comments in the attached pdf.

Best regards

Reviewer 2 Report

  • The abstract is weak. Explain the research problem you have solved and why such a study is needed. The novelty part is missing. Explain what new you have done, which is not done in the literature. Also, the paper aim is misleading, last line says focus is on brazil, but above results are for north America, Europe and Africa.
  • Introduction needs improvement.
    • A literature review was first performed to 67 identify where the majority of articles studying the relationship between fire and water 68 quality are located……But the literature review is not good enough. A more exhaustive literature review is required.
    • we developed a Fire Impact on Water Resources Index (FIWRI)…. You need to explain why this is needed. Is there any similar index available. If yes, what are their limitations, how are you going to improve and why is your index better?
    • Overall, the introduction is very weak, and does-not highlight the background of the study.
    • If paper is focusing on Brazil, then literature should be focused on brazil. You seem to jump between different countries, and then saying very little about brazil fires and water, which should be in your introduction.
  • Materials and methods
    • How is 2.1 LR is a part of methods and literature. Also we are more interested in what literature says, findings and limitations. What words you used for doing your literature is not required, nor readers will be interested. Please do a proper LR and add to introduction or a separate section.
    • Line 95: We extracted the study 95 area of each paper and presented them on a World Map………… (Site the map here, figure number??)
    • Fire Impact on Water Resources Index (FIWRI). Good idea here. I liked the approach. Is there any other fire-water index. You need to mention them. Why is your better. A small para would be beneficial.
    • You have used different data sets. Please provide a summary table for all data used and their specifications.
  • Results
    • I understand you are trying to understand fire-water affects globally, so that you can understand the dominating parameters. But this review could be your another review article. At present you are mixing two different concepts. I would suggest to focus your literature here on brazil, and do experiments on brazil.
    • Fig 5. Improve resolution. Specially lat-long coordinates. Fig title should be standalone. Please specify is this fire avg over 2003-2020 or sum.
    • Fig 8. Improve resolution.
  • Discussion
    • More detailed discussion about the index you proposed is needed.
    • Explain figure 7 results in the discussion section in detail
  • Conclusions
    • Quantify your conclusions based on your results obtained.